# Formulation and Evaluation of Hydrophilic Polymer Based Methotrexate Patches: In Vitro and In Vivo Characterization

**DOI:** 10.3390/polym14071310

**Published:** 2022-03-24

**Authors:** Muhammad Shahid Latif, Fatemah F. Al-Harbi, Asif Nawaz, Sheikh Abdur Rashid, Arshad Farid, Mohammad Al Mohaini, Abdulkhaliq J. Alsalman, Maitham A. Al Hawaj, Yousef N. Alhashem

**Affiliations:** 1Advanced Drug Delivery Lab, Gomal Centre of Pharmaceutical Sciences, Faculty of Pharmacy, Gomal University, Dera Ismail Khan 29050, Pakistan; shahidlatif1710@gmail.com (M.S.L.); sheikhabdurrashid11@gmail.com (S.A.R.); 2Department of Physics, College of Science, Princess Nourah bint Abdulrahman University, P.O. Box 84428, Riyadh 11671, Saudi Arabia; ffalharbi@pnu.edu.sa; 3Gomal Center of Biochemistry and Biotechnology, Gomal University, Dera Ismail Khan 29050, Pakistan; arshadfarid@gu.edu.pk; 4Basic Sciences Department, College of Applied Medical Sciences, King Saud bin Abdulaziz University for Health Sciences, Alahsa 31982, Saudi Arabia; mohainim@ksau-hs.edu.sa; 5King Abdullah International Medical Research Center, Alahsa 31982, Saudi Arabia; 6Department of Clinical Pharmacy, Faculty of Pharmacy, Northern Border University, Rafha 91911, Saudi Arabia; kaliqs@gmail.com; 7Department of Pharmacy Practice, College of Clinical Pharmacy, King Faisal University, Ahsa 31982, Saudi Arabia; hawaj@kfu.edu.sa; 8Clinical Laboratory Sciences Department, Mohammed Al-Mana College for Medical Sciences, Dammam 34222, Saudi Arabia; yousefa@machs.edu.sa

**Keywords:** sodium carboxymethylcellulose (CMC-Na), hydroxypropyl methylcellulose (HPMC), transdermal drug deliveries (TDDs), methotrexate, transdermal patches

## Abstract

This study attempted to develop and evaluate controlled-release matrix-type transdermal patches with different ratios of hydrophilic polymers (sodium carboxymethylcellulose and hydroxypropyl methylcellulose) for the local delivery of methotrexate. Transdermal patches were formulated by employing a solvent casting technique using blends of sodium carboxymethylcellulose (CMC-Na) and hydroxypropylmethylcellulose (HPMC) polymers as rate-controlling agents. The F1 formulated patch served as the control formulation with a 1:1 polymer concentration. The F9 formulation served as our optimized formulation due to suitable physicochemical properties yielded through the combination of CMC-Na and HPMC (5:1). Drug excipient compatibilities (ATR-FTIR) were performed as a preformulation study. The ATR-FTIR study depicted great compatibility between the drug and the polymers. Physicochemical parameters, kinetic modeling, in vitro drug release, ex vivo drug permeation, skin drug retention, and in vivo studies were also carried out for the formulated patches. The formulated patches exhibited a clear, smooth, elastic nature with good weight uniformity, % moisture uptake, drug content, and thickness. Physicochemical characterization revealed folding endurance ranging from 62 ± 2.21 to 78 ± 1.54, tensile strength from 9.42 ± 0.52 to 12.32 ± 0.72, % swelling index from 37.16 ± 0.17 to 76.24 ± 1.37, and % drug content from 93.57 ± 5.34 to 98.19 ± 1.56. An increase in the concentration of the CMC-Na polymer (F9) resulted in increased drug release from the formulated transdermal patches. Similarly, drug permeation and retention were found to be higher in the F9 formulation compared to the other formulations (F1–F8). A drug retention analysis revealed that the F9 formulation exhibited 13.43% drug retention in the deep layers of the skin compared to other formulations (F1–F8). The stability study indicated that, during the study period of 60 days, no significant changes in the drug content and physical characteristics were found. ATR-FTIR analysis of rabbit skin samples treated with the formulated transdermal patches revealed that hydrophilic polymers mainly affect the skin proteins (ceramide and keratins). A pharmacokinetic profile revealed C_max_ was 1.77.38 ng/mL, T_max_ was 12 h, and t_1/2_ was 17.3 ± 2.21. In vivo studies showed that the skin drug retention of F9 was higher compared to the drug solution. These findings reinforce that methotrexate-based patches can possibly be used for the management of psoriasis. This study can reasonably conclude that methotrexate transdermal matrix-type patches with CMC-Na and HPMC polymers at different concentrations effectively sustain drug release with prime permeation profiles and better bioavailability. Therefore, these formulated patches can be employed for the potential management of topical diseases, such as psoriasis.

## 1. Introduction

Skin has widely been used for topical (dermal) delivery, where the drug is localized in layers of the skin and reaches the blood circulation [1].

Skin is protective in nature and acts as a major barrier between the body and the external environment [2]. The first mechanical barrier of the skin is the stratum corneum, which protects the skin from the environment. A majority of the drugs that undergo passive diffusion receive great concern for their absorption process [3]. A series of specialized skin cell layers connected to the stratum corneum are shedding continuously. The rigid layer of skin is the stratum corneum [4].

The nature of the route is of utmost importance for overcoming the stratum corneum’s barrier nature. Numerous techniques, such as lipid vesicles (liposomes, lipophilic lipid conjugates, and iontophoresis), have been used for enhancing topical drug delivery [5].

Psoriasis is a chronic autoimmune skin and joint manifestation disease that is characterized by erythematous plaques producing red and white scales on the epidermis layer of the skin. This disease mainly occurs at the surface of the sacral, scalp, knee, and elbow areas. Psoriasis occurs when the immune system mistakenly sends pathogen signals that facilitate the division and growth of skin cells and result in psoriasis [6].

The first line of treatment protocol for the management of psoriasis has proven to be less effective due to associated toxicities, inconvenient long-term use of medication, cosmetic unacceptability, and poor penetration of the dosage form [7].

Methotrexate is an anti-cancerous drug used for the management of autoimmune diseases and various cancer therapies [8]. Methotrexate is used for the management of mild- to severe-type psoriasis in doses of 10 to 25 mg per week [9].

Methotrexate causes inhibition of the dihydrofolate reductase enzyme and results in the inhibition of cell proliferation. The drug suppresses the autoimmune system and regulates the proliferation of undesired cells [10]. Regardless of undeniable methotrexate advantages, some disadvantages have also been reported, such as unpredictable bioavailability, poor absorption, and a short half-life. The use of a higher dose showed no specific toxicities on proliferating normal cells [11].

These problems can be overcome using a topical administration of methotrexate. The topical use of methotrexate offers numerous advantages over injection and oral routes of administration. This route is noninvasive, painless, convenient, and offers quick therapy termination in the case of development of any adverse effects [12]. Transdermal drug delivery offers controlled drug delivery for a longer time period, and this route of drug administration avoids systemic first pass and presystemic metabolism [13].

Numerous formulations, such as ufasomes, solid-in-oil nanocarriers, passive and iontophoretic transdermal deliveries, nanostructured lipid carriers (NLCs), nanogels, various surfactants with hydrophilic gels, deformable liposomes, liposomes, and transcutaneous deliveries, have been used for achieving greater efficacy and minimizing side effects associated with the use of methotrexate. Inspite of huge efforts, no efficacious or effective topical dosage forms are available yet commercially [14].

Skin drug delivery is an alternative route for drugs with low bioavailability. A transdermal patch is placed on the skin, and the drug reaches the systemic circulation system following a controlled-release mechanism [12].

In the preparation of transdermal patches, polymers play a crucial role in achieving mechanical properties, as well as controlled drug release from patches. For the development of proper transdermal drug delivery systems, suitable matrices are necessary [15]. The application of polymers in transdermal drug delivery ensures controlled drug release, permeation, and targeted drug delivery systems. Polymers offer controlled drug release from transdermal patches [16].

CMC-Na is anionic derivative of carboxymethylcellulose. It provides swelling properties and uniform viscosity, and it is resistant to bacterial decomposition. It produces a film-like structure on the surface of the skin, prevents skin moisture, and has extensively been used in topical pharmaceutical formulation for its controlled and sustained targeted drug delivery [17].

Hydroxypropyl methylcellulose (HPMC) is a derivative of cellulose ether. It is a nontoxic, biocompatible, and biodegradable polymer with numerous applications in controlled drug delivery, cosmetics, and adhesive in formulations. The thickening, gelling, and swelling properties of HPMC make its use convenient for the formulation of controlled or sustained drug delivery systems [17].

This study aimed to formulate controlled-released transdermal patches of methotrexate using hydrophilic polymers (CMC-Na and HPMC) at different concentrations with a plasticizer (PEG400) to offer maximum skin release and permeation, as well as localization of the effective drug in the deep layers of the skin to attain more clinical effectiveness compared to systemic toxicity. Our study showed that these polymers have outstanding transdermal-patch-forming ability. Albino rabbit skin was used for investigating an in vitro drug release study and ex vivo drug permeation study, and methotrexate retention studies in the deeper skin layers. The formulated transdermal patches were evaluated for physicochemical properties. ATR-FTIR spectroscopy was used for assessing the molecular-level skin interactions of the drug and the polymers.

## 2. Materials and Methods

### 2.1. Materials

Methotrexate was obtained from Wilson’s Pharmaceuticals Pvt. Ltd., Pakistan. Ethanol, PEG400, sodiumhydroxide (NaOH) (Sigma-Aldrich, P.O. Box 14508, St. Louis, MO, USA, +1-314-771-5765, Sigma-AldrichChemie GmbH, Riedstr, 2, 89555 Steinheim, Germany, +49-7329-970), dichloromethane, calcium chloride (The Dow Chemical Company., 693 Washington St. #627, Midland, MI 48640, USA), and distilled water were used in the preparation of the buffers and patches. Hydroxypropyl methylcellulose (HPMC) and sodium CMC (CMC-Na) (Sigma-Aldrich, P.O. Box 14508, St. Louis, MO, USA, +1-314-771-5765, Sigma-AldrichChemie GmbH, Riedstr, 2, 89555 Steinheim, Germany, +49-7329-970), were used as rate-controlling agents.

### 2.2. Formulation of Transdermal Patches of Methotrexate

Transdermal patches of methotrexate were prepared by a solvent casting technique. At different concentrations, the polymers CMC-Na and HPMC were also employed in the fabrication of transdermal patches as rate-controlling polymers. Measurements of the drug and the polymers were completed using a Shimadzu AX200 analytical balance (Nishinokyo-Kuwabara-cho, Nakagyo-ku, Kyoto 604-8511, Japan, +81-75-823-1111). The weighed polymers (CMC-Na and HPMC) were then dissolved in an equal volume (10 mL) of distilled water and ethanol. The solution was continuously stirred for a time period of 30 min using a magnetic stirrer set at 500 rpm. A specified amount of methotrexate was dissolved in 10 mL phosphate buffer solution (pH 7.4). The literature revealed that methotrexate is hydrosoluble at physiological pH, so a phosphate buffer solution of pH 7.4 was employed to dissolve the methotrexate [18]. The mixture of polymer solvents was added to the drug aqueous solution with continuous stirring for uniform mixing and distribution. For the prevention of brittleness upon storage, PEG400 was added to the polymeric patches.

A magnetic stirrer (500 rpm) was employed for the final dispersion process. In order to remove the entrapped air bubbles, the solution was placed in a D-78224 Singen Sonicator (ElmaSchmidbauer, Gottlieb-Daimler-Straße 17, Hohentwiel, Germany, +49-7731-8820) for 15 min and then poured into glass petri dishes with 19.5 cm^2^ surface areas. The glass petri dishes were placed in an oven at 40 °C for 12 h. The dried patches were removed from the glass petri dishes carefully Table 1. The patches were folded into aluminum foil and placed in a desiccator until further study. The formulation optimization was achieved by varying the concentrations of CMC-Na and HPMC. The 5:1 polymeric blend of CMC-Na and HPMC provided optimum physicochemical profiles, such as higher values for folding endurance, % swelling index, tensile strength, and % drug content, as mentioned in Table 2.

### 2.3. Compatibility Studies (Preformulation Study)

Drug and polymer compatibility studies were carried out to evaluate any incompatibilities. An ATR-FTIR study was performed on methotrexate (pure drug) and its mixtures in the transdermal patches (F1–F9). A total of 32 scans were obtained at resolutions ranging between 4000 and 600 cm^−1^ for evaluating each spectrum.

### 2.4. Physicochemical Evaluation of MTX Loaded Patches

The following parameters were evaluated for the prepared transdermal patches.

#### 2.4.1. Surface pH

An InoLab^®^pH meter (Xylem Analytics, Dr. Karl Slevogt Street 1. Weilheim 82362, Germany) was utilized for the evaluation of surface pH of the formulated patches. In a test tube, 1ml of distilled water and a 1 cm^2^ portion of transdermal patch was kept at room temperature (25 ± 2 °C) for at least 2 h. A filtration process was then employed for removing excess water from the test tube. The pH meter was placed at three different places at the swollen part of the patch for calculating the average (mean ± SD) result [16].

#### 2.4.2. Physical Appearance

The formulated patches were evaluated for homogeneity, transparency, clarity, color, and smoothness.

#### 2.4.3. Measurement of Thickness

Thickness of the formulated transdermal patches was evaluated using a vernier caliper (Ossenpaß 4, 47623 Kevelaer, Germany, +49-2832-92390). At six different places, the thickness of the formulated transdermal patches was measured, and the results were calculated and averaged [19].

#### 2.4.4. Weight Variation

Weight variations between the formulated transdermal patches were evaluated using a Shimadzu AX200analytical weighing balance (Nishinokyo-Kuwabara-cho, Nakagyo-ku, Kyoto 604-8511, Japan, +81-75-823-1111). Each transdermal patch was individually weighed, and then the weight of each individual patch was checked in contrast with the formulated patch average weight [20].

#### 2.4.5. Folding Endurance

The folding endurance of patches is of utmost importance for the determination of the plasticizer and its nature. The formulated patches were folded constantly at the same place until a crack or break appeared. The folding of patches at the same place provided the value for folding endurance [19].

#### 2.4.6. Tensile Strength (Kg/cm^2^)

The determination of the mechanical properties of the formulated transdermal patches was conducted using a pulley apparatus. The initial patch length was identified using a scale. One side of the transdermal patch was attached to a weighing balance hook, and the other side was attached toa rope that crossed over the pulley and attached to a weighing pan. In the pan, weight gradually increased until a crack or break appeared in the patch. Tensile strength was calculated by the total weight present in the pan. The percent elongation of work completed was evaluated using a thread pointer.

The following equation was used for the identification of force required for breaking or cracking of the transdermal patch [21]:(1)Tensile Strength=F / [a ∗ b (1+L / I)]
where *F* is the force required to break the patch, *a* is the width of the patch, and *b* is the thickness of the patch (cm). *L* is the length of the patch (cm), and *I* is the elongation of the patch (cm) before breaking or cracking occurred.

The percent elongation of the formulated transdermal patches was evaluated using this equation:(2)% Elongation=(Lf−Li) / Li ∗ 100
where *Lf* is the length of the patch before breaking, and *Li* is the initial length of the patch.

#### 2.4.7. Percent Moisture Content (%)

The individual weight of all formulated transdermal patches was obtained for evaluation of the percent moisture content. In a desiccator, silica was placed with the formulated patches for 24 h. Constant weights of the patches were evaluated by reweighing each patch. The difference in the weights of the formulated patches provided the value of percent moisture content [22].

The following equation was used for the calculation of percent moisture content:(3)% Moisture Content=(wi−wf) / wi ∗ 100
where *wi* is the initial weight of the patch, and *wf* is the final weight of the patch.

#### 2.4.8. Evaluation of Swelling Index

The formulated patches were weighed accurately for determination of the percent swelling index. Aluminum chloride and the patches were placed in a desiccator to maintain humid conditions. After 3 days, the patches were removed from the desiccator. The patches were weighed again. The difference between the initial and final weights of the patches provided the value of the percent swelling index. Finally, the average percent swelling index was calculated. The amount of water absorbed per unit weight in accordance with the undissolved patches provided the value of swelling index [23]:(4)SI=(w2−w3) / w3×100
where *w*2 is the swollen patch weight, and *w*3 is the patch weight after placement in a desiccator. Triplicate readings were tabulated as the mean.

#### 2.4.9. Estimation of Drug Content

Drug content was estimated for the transdermal patches. Volumetric flasks were used for the identification of drug content. Volumetric flasks were filled with phosphate buffer (pH 7.4), and then the formulated transdermal patches were placed in the volumetric flasks, which were placed in a sonicator for a time period of 6 h. Then, the solution was filtered. The drug content was evaluated using a UV-visible Shimadzu AX200spectrophotometer (Nishinokyo-Kuwabara-cho, Nakagyo-ku, Kyoto 604-8511, Japan, +81-75-823-1111) set at 303 nm wavelength [16].

#### 2.4.10. Water Vapor Transmission Rate

Glass vials of transmission cells were dried completely by keeping in an oven. An amount of 1 g of anhydrous calcium chloride was placed in transmission cells. Transdermal patches were weighed individually and fixed at the brim. The transmission cells were placedin a desiccator. A solution of potassium chloride was placed in the desiccator for maintaining 84% humidity. The transmission cells were removed after specific intervals of time, i.e., 6, 12, 24, 36, 48, and 72 h, and were weighed again after completion of the study [24].

### 2.5. Stability Study

Stability studies for the formulated transdermal patches were completed for a time period of 60 days. Incubation of the transdermal formulated patches was conducted at 25 ± 2 °C with 60 ± 5% RH and 40 ± 2 °C with 75 ± 5% RH. The physical appearance and drug content of the formulated patches were evaluated at a regular interval of 20 days. Triplicate readings were tabulated as the mean ± SD (*n* = 3). Estimation of drug content was completed using the procedure previously discussed in Section 2.4.9.

### 2.6. Skin Irritation Study

For the skin irritation study, proper NOC from the research center was obtained. Healthy male albino rabbits of 2–2.5 kg weight was used in this study. Published and standard food was provided to the rabbits at least three days before starting the experiment. The food was comprised of 40% bran, 20% grass meal, 18% middlings, and 10% white fish meat; water was also added ad libitum [25]. The temperature was maintained at 25 ± 2 °C with relative humidity controlled at 50 ± 10% for all rabbits. An electrical clipper was used for shaving sparse hairs from the rabbits’ abdomens. The shaved areas were cleaned with dry cotton. For evaluating skin irritation studies, the Draize patch test was used. Rabbits were divided into five different groups, with each group containing 3 rabbits. The skin irritation study was completed in 1 week. The nontreated group served as Group1; the USP adhesive tape group (control group) served as Group2; the methotrexate-loaded patch group served as Group3; the 0.8% *v*/*v* aqueous solution of the standard irritant formalin group served as Group4; and the blank formulated patch group served as Group5. The grading of skin irritation was completed using a visual scale. No skin irritation was indicated with “0”, slight skin irritation was indicated with “1”, well-identified skin irritation was indicated with “2”, moderate skin irritation was identified with “3”, and formulation of a scar on the skin was indicated with “4” [16].

### 2.7. In Vitro Drug Release Study

An in vitro drug release study for the methotrexate-loaded transdermal patches was completed using a Franz Diffusion Cell Apparatus (PermeGear, 1815 Leithsville Rd, Hellertown, PA, 18055, USA, +1 484-851-3688). For the in vitro drug release study of the methotrexate-loaded transdermal patches, a Tuffryn membrane was used. The Tuffryn membrane was placed on the diffusion cells’ and a 1cm^2^ area of transdermal patch was fixed over the Tuffryn membrane. Receptor compartments were filled with phosphate buffer (pH 5.5). The temperature of the receptor compartments was maintained at 32 ± 0.5 °C with constant stirring at 100 rpm using the beads of a magnetic stirrer. At specific time intervals of 0.5, 1, 1.5, 2, 4, 8, 12, 16, and 24 h, 2 mL samples were collected in test tubes and replaced with fresh phosphate buffer (pH 5.5) at equal volume for maintaining sink conditions. The collected samples were analyzed spectrophotometrically at 303 nm wavelength [26].

### 2.8. Kinetics of Drug Release

The following kinetic models were used for the evaluation of drug release kinetics from methotrexate-loaded transdermal patches [27,28].

#### 2.8.1. Zero Order Kinetics

This model was used for the identification of sustained drug release. This model was also used for the identification of constant release rate of the drug, which does not disintegrate from a dosage form.

The following equation was used for zero order kinetics:(5)$$W=k_{1\;}t$$
where *W* is the drug released with respect to time (*t*), *k*_1_ is the constant for zero order kinetics, and *t* is time.

#### 2.8.2. First Order Kinetics

This model was used in conditions where the sink condition existed. First order kinetics were used for the evaluation of absorption, as well as the elimination rate of the drug from the system.

The following equation was used for first order kinetics:(6)$$ln\;\left(100-W\right)=ln\;100-k_{2}t$$
where *W* is the drug released with respect to time (*t*), *k*_2_is the constant for first order kinetics, and *t* is the total time.

#### 2.8.3. Hixon Crowell Model

The Hixon Crowell model was first proposed by Hixon and Crowell. This model is also known as the ‘Erosion Model’.

The following equation was used in the Hixon Crowell model:(7)$$\left(100-W\right)\;1\;/\;3=100\;\left(1\;/\;3\right)-k_{3}t$$
where *W* is the released drug with respect to time (*t*), *k*_3_*t* shows the relationship between surface volume and the constant for Hixon Crowell, and *t* is the total time.

#### 2.8.4. Higuchi Model

The Higuchi model is also known as the ‘Diffusion Model’. The Higuchi model is used for determination of the dissolution rates of certain dosages that are not ointments. This model is also used for the evaluation of non-erodible matrices, such as ointment bases.

The following equation was used in the Higuchi model:(8)$$W=k_{4}t^{1/2}$$
where *W* is the released drug with respect to time(*t*), *k*_4_ is the constant for the Higuchi dissolution rate, and *t* is the total time.

#### 2.8.5. Power Law Equation

This equation is also known as the Korsmeyer–Peppas equation. The following equation expresses the power law equation:(9)$$M_{t}\;/\;M_{\infty }=k_{4}\;t^{n}$$
where *M_t_*/*M*_∞_ is the released amount of drug with respect to time (*t*), *k*_4_ is the constant for the power law equation, and *n* is the exponent of the diffusion that describes the transport behavior of drug.

The value of *n* depicts the drug release mechanism. The mechanism of drug release is said to be Fickian diffusion when the *n* value is equal to 0.45. When the value of *n* is equal to 0.45 and less than 0.89, the drug release mechanism is said to be non-Fickian, or anomalous, diffusion. When the value of *n* is equal to 0.89, the drug release mechanism is said to be Case II transport, and when the value of *n* is greater than 0.89, the drug release mechanism is known as Super Case II transport.

### 2.9. The Ex Vivo Permeation Study

#### Preparation of Rabbit Skin

For the ex vivo permeation study, NOC from the GCPS center was taken. In this study, healthy male albino rabbits of 2–2.5 kg were used. Published and standard foods were given to the rabbits at least three days before starting the experiment. The food was comprised of 40% bran, 20% grass meal, 18% middling, and 10% white fish meat; water was also added ad libitum [25]. The temperature was maintained at 25 ± 2 °C with relative humidity controlled at 50 ± 10% for all the rabbits. Rabbits were euthanized by an overdose of xylazine and ketamine. The excised skins were placed in warm water for 30s to remove adhered fats. The skins were washed with a sodium chloride solution (0.9% *w*/*v*), and the skins were cut to the appropriate size. The skins were then placed in freezer at −20 ± 1 °C until further use [29].

For ex vivo permeation studies, a Franz Diffusion Cell Apparatus was used. Receptor fluids were filled with phosphate buffer (pH 7.4). The temperature of the receptor medium was regulated at 37 ± 0.1 °C, and the medium was stirred constantly with magnetic beads at 150 rpm. Phosphate buffer (pH 7.4) at 20 °C was used for hydration of the skins. The skins were placed on the diffusion cells, and a1cm^2^ piece of transdermal patch was fixed over the rabbit skin. At predetermined intervals of time (0.5, 1, 1.5, 2, 4, 8, 12, 16, and 24 h), samples of 2mL were collected in test tubes, and fresh phosphate buffer (pH 7.4) was replaced with an equal volume to maintain sink conditions. The collected samples were analyzed spectrophotometrically at a 303 nm wavelength [29].

### 2.10. Drug Retention Study

After the completion of ex vivo drug permeation studies, drug retention studies were carried out. The skins were removed carefully from the Franz Cell Apparatus and washed with normal saline. For epidermis–dermis separation, the skin was cut in correspondence of the permeation area, and the skin was heated at 50 °C with a hair dryer for 30 s; then, the epidermis was separated from the dermis by scraping with a scalpel. Epidermis and dermis samples were placed in glass vials and the drug was extracted with methanol. A UV-visible spectrophotometer was used for analyzing samples at a 303 nm wavelength. The triplicate values were carried out, and the results were averaged.

### 2.11. Skin ATR-FTIR Spectroscopy

After the completion of the ex vivo permeation study, the skins were carefully removed from the diffusion cells. The skins were washed with a phosphate buffer solution (pH 7.4) and blotted with dry, soft tissue. The skins were placed onto the internal reflection element of an ATR-FTIR spectrometer. A force gauge of 80 N was applied on the top of each sample to ensure reproducible contact between the crystal and the samples. Normalization of the skin was conducted to minimize inter-sample variation. ATR-FTIR spectra were obtained in the frequency range from 4000 to 650 cm^−1^ with a spectrum resolution of 4 cm^−1^. Perkin Elmer Spectrum Version 6.0.2 software was used for assigning peak positions.

### 2.12. In Vivo Studies

Male healthy albino rabbits of 2–2.5 kg were used in this study. Standard food was given to the rabbits for a period of 7 days. The food was comprised of 40% bran, 20% grass meal, 18% middlings, and 10% white fish meat; water was also added ad libitum [25]. The temperature was maintained at 25 ± 2 °C with relative humidity controlled at 50 ± 10% for all the rabbits. For the ex vivo permeation study, NOC from the GCPS center was obtained, and all the procedures followed international standards and ethical norms. The rabbits were anesthetized using a ketamine-xylazine injection (I/M). An electric clipper was used for shaving the backs of the rabbits in an area of 5 cm and was cleaned with an ethanol swab.

The rabbits were separated into two groups; each group was comprised of 6 rabbits. An aqueous solution of methotrexate was administered to Group-A, which served as the control group. The optimized patch formulation F9 was applied to Group-B, which served as the experimental group. At predetermined time intervals, 0.5 mL of blood was collected and centrifuged for the separation of plasma. Methanol was added to the plasma and vortexed for 10 min, and then it was centrifuged at 5000 rpm for 3 min. Supernatant was added to the filtration and was dissolved in 0.5 mL HPLC mobile phase pH 6 (0.2M Na_2_HPO_4_ and 0.2M citric acid; 2:1), and acetonitrile in 90:10 *v*/*v* was used for the evaluation of HPLC to estimate the drug plasma content.

After the completion of the permeation study, the rabbits were euthanized by the administration of an overdose of ketamine–xylazine injection. Meticulously, the excised skin was cut into small pieces and washed with a saline solution. The skin was dipped in distilled water for 24 h in order to remove methotrexate and filtered, and HPLC was used for the determination of residual drug concentrations [29].

### 2.13. Statistical Analysis

The statistical tools used in this study were Student’s *t*-test and one-way ANOVA/post hoc analysis using Tukey’s honestly significant difference test. SPSS version 16 software was used for statistical analysis. *p* < 0.05 was considered as significant. Triplicate values were observed, and the results were expressed as the mean value ± standard deviation.

## 3. Results

This study reflected an important strategy to develop methotrexate-containing transdermal patches with different ratios of hydrophilic polymers (CMC-Na and HPMC). The formulated patches were evaluated with release, permeation, and skin drug retention studies. Furthermore, an investigation of hydrophilic polymers on the skin was also conducted.

### 3.1. Drug Excipients Compatibility Studies (Preformulation Study)

The ATR-FTIR spectrum of pure methotrexate showed absorption characteristic bands at 3450 cm^−1^ (O–H stretching from carboxyl groups).The bands appeared at 3080 cm^−1^ were allocated to (N–H stretched primary amine), 1670, and 1600 cm^−1^ were allocated to stretched C=O (the –C=O stretched carboxylic group and the C=O stretched amide group). As shown in Figure 1, the bands of the methotrexate sample split into two C=O groups. At 1550–1500 cm^−1^, the bands corresponding to the bends of N–H formed an amide group, and overlapping of spectral ranges with aromatic –C=C stretching occurred. Additional major bands, such as 1400–1200 cm^−1^, corresponded to –C–O stretching from carboxylic groups; 930 cm^−1^ corresponded to O–H bends out of planes, and 820 cm^−1^ corresponded to CH_2_–adjacent hydrogen on aromatic rings. An ATR-FTIR study depicted that major peaks of the drug and polymers were found preserved, which confirms the purity. Pure drug methotrexate spectra and spectra of physical mixtures with varying ratios of CMC-Na and HPMC (F1 to F9) showed the presence of principal peaks within the formulations, eliminating the likelihood of any significant interactions. The ATR-FTIR study depicted that methotrexate and polymer major peaks observed in the formulations were preserved.

### 3.2. Physicochemical Assessment of Methotrexate-Loaded Transdermal Patches

The formulated patches (F1–F9) were prepared using different concentrations of CMC-Na and HPMC, as shown in Figure 1. Prepared patches were subjected to visual inspection to analyze physical appearance. The physical appearance gave satisfactory results. All prepared patches were found to be smooth, non-sticky, opaque, homogeneous, and flexible in nature (Table 1). The formulated transdermal patches (F1–F9) exhibited pH levels in the acceptable range i.e., 5.2 to 5.9; hence, the transdermal patches are suitable for use, and no skin irritation will occur. The transdermal patch thickness ranged from 0.51 ± 0.02 mm to 0.87 ± 0.06 mm. This thickness variation could be attributed to the nature and concentrations of hydrophilic polymers, i.e., the increase in CMC-Na polymer concentration resulted in an increase in the patch thickness. The thickness data obtained from this study were matched with the specified range of thickness in the literatures review for transdermal patches [30].

**Table 1 polymers-14-01310-t001:** Composition of methotrexate transdermal patches.

		Total Amount of Polymers				Amount of Solvents (*v*/*v*) mL	
Batch	Amount of MTX (mg)	CMC-Na (mg)	HPMC (mg)	Combination CMC-Na/HPMC	Plasticizer PEG400 (%)	Ethanol	Distilled Water
F1 (Control)	5	100	100	1:1	30	10	10
F2	5	100	200	1:2	30	10	10
F3	5	100	300	1:3	30	10	10
F4	5	100	400	1:4	30	10	10
F5	5	100	500	1:5	30	10	10
F6	5	200	100	2:1	30	10	10
F7	5	300	100	3:1	30	10	10
F8	5	400	100	4:1	30	10	10
F9	5	500	100	5:1	30	10	10

The formulated transdermal patches exhibited weight variations between 73.86 ± 0.025 mg and 103.33 ± 1.150 mg. The study revealed that the increase in the concentration of CMC-Na resulted in an increased weight of patches compared to HPMC. This might be due to CMC-Na possessing a greater affinity for water and greater moisture uptake, leading to increased patch weight compared to HPMC. The CMC-Na polymer is hygroscopic in nature; it caused retention of water in the patches and resulted in increased weight of patches, while HPMC formulated a thin patch matrix [31]. Folding endurance is of utmost importance for patches because greater folding endurance prevents patches from being easily broken or damaged, and patches are considered to meet good quality [32]. All the formulated patches (F1–F9) showed greater folding endurance (>60 times). This showed that all transdermal patches met the standard patch requirements. Different concentrations of the polymers (HPMC and CMC-Na) did not affect the folding endurance value for the formulated transdermal patches. PEG400 was used as a plasticizer for flexible patch formulation. The formulated transdermal patches depicted tensile strength and elongation values ranged between 9.42 ± 0.52 kg/cm^2^ and 12.33 ± 0.72 kg/cm^2^. From the obtained data, it was cleared that all the formulations had sufficient tensile strength, and the variations were within the acceptable limit (Table 2).

**Table 2 polymers-14-01310-t002:** Characterization of transdermal patches of methotrexate (F1–F9).

Characteristics									
F Codes	pH	Thickness (mm)	Weight Variation (mg)	Folding Endurance	Tensile Strength kg/cm^2^	% Moisture Content	% Swelling Index	% Drug Content	Water Transmission Rateg/m^2^/24 h
F1	5.2	0.51 ± 0.02	73.86 ± 0.03	62 ± 2.21	9.62 ± 0.43	9.26 ± 1.52	37.16 ± 0.17	95.75 ± 4.22	3.23 ± 0.15
F2	5.4	0.59 ± 0.03	75.34 ± 0.04	64 ± 3.14	10.37 ± 0.26	10.21 ± 1.16	42.65 ± 1.46	96.80 ± 3.32	3.34 ± 0.26
F3	5.4	0.63 ± 0.01	77.88 ± 0.03	63 ± 1.43	12.32 ± 0.47	10.27 ± 1.25	47.43 ± 0.58	93.57 ± 5.34	3.68 ± 0.38
F4	5.6	0.72 ± 0.05	79.75 ± 0.03	67 ± 3.23	11.72 ± 0.57	10.35 ± 1.45	57.02 ± 0.25	97.50 ± 3.21	3.92 ± 0.29
F5	5.9	0.73 ± 0.06	80.33 ± 0.04	69 ± 2.55	9.42 ± 0.52	10.89 ± 1.71	62.58 ± 1.89	94.15 ± 4.23	4.15 ± 0.33
F6	5.7	0.62 ± 0.03	78.54 ± 0.05	71 ± 1.34	10.25 ± 0.92	10.85 ± 1.50	48.73 ± 0.54	97.12 ± 2.43	4.27 ± 0.46
F7	5.4	0.69 ± 0.02	87.66 ± 0.03	73 ± 3.21	9.57 ± 0.13	11.25 ± 1.23	57.85 ± 1.15	96.23 ± 3.32	4.48 ± 0.72
F8	5.5	0.78 ± 0.04	96.54 ± 0.21	76 ± 4.23	11.14 ± 0.28	11.63 ± 1.50	65.61 ± 0.21	97.39 ± 2.35	4.79 ± 0.64
F9	5.7	0.87 ± 0.06	103.33 ± 0.15	78 ± 1.54	12.33 ± 0.72	11.85 ± 1.56	76.24 ± 1.37	98.19 ± 1.56	4.92 ± 0.78

Data are expressed as mean ± SD; *n* = 3; *p* > 0.05.

The moisture content ranged from 9.26 ± 1.5% to 11.85 ± 1.56%. In the prepared formulations, the moisture content varied. The formulations containing greater amounts of CMC-Na resulted in an increase in moisture content. The CMC-Na polymer is hygroscopic, and it can cause absorption, as well as retention, of water in transdermal patches. It resulted in higher moisture uptake compared to the HPMC polymer [33]. Swelling of the patches was observed in distilled water, and an increase in weight was observed due to the swelling, which was the highest for formulation F9, containing a combination of a higher amount of CMC-NA and a lower amount of HPMC. F1 exhibited the least swelling due to the presence of a low and equal amount of the HPMC and CMCNa polymers. The F6 to F9 formulations contained higher amounts of CMC-Na with respect to the HPMC polymer and exhibited an increase in the swelling index. Hence, it can be concluded that the concentration of the CMC-Na polymer played a major role in the increased swelling index (Table 2). The formulated patches showed uniform drug content analysis and demonstrated a minimum variability of batch.

This study indicated that formulated patches were capable of providing uniform drug content analysis. The formulated patches exhibited drug content uniformity within pharmacopeial limits. The drug content ranged from 93.57 ± 5.34 to 98.19 ± 1.56. This drug content range is convenient for transdermal application. The transdermal formulated patches showed a water vapor transmission rate (WVTR) ranging from 3.23 ± 0.15 to 4.92 ± 0.78 (Table 2). This test was performed to verify transdermal patch permeability characteristics. It is pertinent that low values of WVTR can significantly impart formulation stability during long-term storage. This study revealed that an increase in CMC-Na concentration resulted in an increased rate of WVTR. The F1 formulation (CMC-Na:HPMC of 1:5) showed the smallest WVTR value, and the F9 formulation (CMC-Na:HPMC of 5:1) showed the greatest WVTR value. This might be due to CMC-Na’s hydrophilic nature. Although no statistically significant difference was found in the aforementioned physicochemical attributes of methotrexate transdermal patches in formulations F1–F9 (*p* > 0.05), due to excellent physicochemical attributes, the formulation F9 with a 5:1 ratio of CMC-Na and HPMC was considered an optimized formulation and was selected for in vivo analysis.

### 3.3. Stability Study

The physical characteristic changes and drug content during the stability study showed no significant changes, as shown in Table 3. The transdermal patches depicted almost the same drug content value observed in the start of the study. The prepared patches exhibited an elastic and flexible nature, and it was observed in the start and at the end of this study. The results of this study demonstrated that all formulated transdermal patches (F1–F9) were found to be chemically and physically stable for a time period of 60 days. There was no significant statistical difference observed when the % drug contents of various formulations of methotrexate transdermal patches stored at ambient, as well as accelerated, temperatures were compared with the initial % drug content (*p* > 0.05).

### 3.4. Skin Irritation Study

During the skin irritation study, no irritation, edema, or erythema was observed for the formulated patches (F1–F9), even after removal of the transdermal patches, which clearly showed that all the formulated transdermal patches were non-irritating. The standard irritant formalin showed severe-type erythema and edema. According to the Draize test, the value for the formulated patches was <2, which indicates that the test was negative and the formulated patches are safe to use transdermally, as shown in Figure 2 [34]. A statistically significant difference was found when formalin (control) was compared to rest of the formulations (F1–F9) in skin erythema and edema scores (*p* < 0.05), as indicated in Figure 2.

### 3.5. In Vitro Drug Release Study

Distribution of the drug must be uniform in the formulated patches, as this ensures release of the drug in a sustained fashion. Drug molecules have greater intermolecular forces compared to those between polymers and the drug. Polymer and drug molecules interact physically via electrostatic movement. HPMC is a cellulose-ether-derived, nontoxic polymer with numerous applications in adhesives, cosmetics, and controlled drug delivery [35]. Its swelling, gelling, and thickening properties make it suitable for designing, formulating tougher and controlled drug delivery systems. CMC-Na is resistant to bacterial decomposition, and it is a derivative of sodium and carboxymethyl cellulose. CMC-Na has widely been used in oral topical formulations, as it causes the prevention of moisture loss and forms a thin layer on the surface of the skin [36]. This study is of utmost importance because it exhibits the behavior of the drug in advance, that is, how this drug will behave in the in vivo studies. Therefore, this study minimized the risk of untoward effects of drugs used directly in living systems. In this study, variable blends of the CMC-Na and HPMC polymers at different concentrations were observed. Transdermal patches offer controlled drug delivery by governing the diffusion method, and the polymers matrixes have greater influence on the diffusivity of motion for small particles, as these particles are limited by chains of the three-dimensional polymer network [37].

The in vitro drug release study depicted increased drug release in the following order: F9 > F8 > F7 > F6 > F5 > F4 > F3 > F2 > F1. Faster drug release was observed from formulated patches containing greater amounts of the hydrophilic polymers. The study also depicted an increase in hydrophilic polymer that resulted in an increase in burst effect, as well as drug release in the formulation. This is attributed to the hydrophilic nature of both polymers (CMC-Na and HPMC), since less time lag was needed to maintain the concentration profiles. Greater thermodynamic activities were observed in the formulated transdermal patches with the use of CMC-Na, as it is hydrophilic in nature and possesses a greater affinity for water [38].

After good initial burst release, the formulations F1 and F2 showed controlled-release profiles through a period of 24 h (Figure 3). The F1 and F2 formulations have smaller particle sizes and showed initial burst release because they offered increased surface area for dissolution. At the end of the 24 h release study, the % drug released from F9 was 92.87% during the same period of study. This burst release is advantageous in chronic case of psoriasis because burst release offers a greater amount of drug release, as well as absorption of the drug into the inflamed lesions of psoriasis [39].

Formulation F1 exhibited the lowest drug release at 67.45%. It was depicted from the study that the increase in the concentration of CMC-Na resulted in increased drug release from the formulated patches. This is because CMC-Na has a greater affinity for water, which results in the uptake of water. Greater thermodynamic activity was observed, and, thus, more amount of the drug was released from the formulated patches.

### 3.6. Drug Release Kinetics

Table 4 describes different parameters of the kinetic models for the identification of methotrexate release patterns from the formulated transdermal patches containing CMC-Na and HPMC polymers as rate-controlling agents. The release data of the formulated patches were fitted to the following models: Korsmeyer model, Hixon–Crowell, Higuchi, first-order kinetics model, and zero-order kinetics model. All formulations exhibited linear relation based on the kinetic models. The drug release data of all the formulations were suitably fitted to the kinetic models, but the percent release data of methotrexate was best fitted to the power law equation. HPMC and CMC-Na at a 1:5 ratio showed a greater “r” value with the best linear relation. By a keen observation of Table 5, it can be observed that release of the drug from all formulated transdermal patches exhibited anomalous, non-Fickian release mechanisms, as the *n* values ranged between 0.517 and 0.843. This indicates that the release of methotrexate from the formulated transdermal patches was achieved by a diffusion method, followed by improved erosion and swelling.

### 3.7. Ex Vivo Drug Permeation Study

This study depicted the rate of absorption, as well as the drug percutaneous absorption mechanism. This study is of utmost importance to minimize the ongoing effects of the drug. Physicochemical characteristics of formulations are responsible for affecting the rate of diffusion. These characteristics include mode of application, surface charge, drug loading capacity, and hydrogen bonding. Ex vivo skin permeation studies depicted that transdermal patch formulation offered a release rate-controlling agent of methotrexate and targeted delivery in the skin. The result of the formulated transdermal patches containing CMC-Na and HPMC at different concentrations depicted in Figure 4, that the F9 formulation showed a greater percent cumulative amount of drug permeation (36.34%), offering significant differences (*p* < 0.05)in contrast with formulation F1 (19.45%) in 24 h. Due to the presence of CMC-Na in a higher concentration in case of the F9 formulation, the stratum corneum experienced positive results in hydration, and the cells of the corneum were allowed to swell, thereby facilitating better passage channels for the drug with wider and greater diffusivity of the drug. Furthermore, the existence of a suitable concentration of surfactants can also alter stratum corneum barrier function by drug penetration from the epidermal layer of the skin to the dermal layer. The F1 formulation produced 14.93 µg/h/cm^2^, which is less than the required flux of 21.09 µg/h/cm^2^. Decreased partitioning of methotrexate from skin into the vehicle was responsible for the decrease in flux value. An increase in CMC-Na concentration amplified the values of flux; the F9 formulation, containing a higher amount of CMC-Na, offered about a 2.5-fold increase in flux (52.72 µg/h/cm^2^) compared to the target flux. This may be due to the interaction of gel layer properties in the higher concentration of CMC-Na with the stratum corneum lipid alkyl chains, resulting in decreased diffusion resistance of the barrier and increased drug penetration into the skin.

### 3.8. Drug Retention

Skin drug retention studies are an important tool for treating numerous local diseases. Methotrexate can effectively produce local action to manage psoriasis if provided in the form of transdermal patches. Our drug retention study revealed that the F9 formulation showed greater drug retention in the deep skin layers compared with other formulations (F1–F8) (Figure 5). This is because the F9 formulation showed greater interaction of the drug with keratinocytes than the other formulation counterparts. It is also supposed that, in the dermal layers, a greater amount of the drug can be retained compared to epidermis because the thickness of the dermis layer is greater than the epidermal layer of the skin. The methotrexate accumulation in the deeper layer of the skin is beneficial in cases of psoriasis, as these deeper layers of skin are mainly affected (one-way ANOVA, *p* < 0.05).

### 3.9. Effect of MTX-Loaded Patch Formulations on Skin Structure

An ATR-FTIR technique was used for the identification of bands with different wave numbers and lipid matrix molecular organization. At different wave numbers, the stratum corneum exhibited different bands in accordance with protein and lipid molecular vibrations. The major peaks of asymmetrical and symmetrical stretched vibrations of CH_2_ were located at about 2918.05 cm^−1^ and 2855.08 cm^−1^, respectively, and derived from chains of lipid hydrocarbons. Furthermore, the bands at about 1650 cm^−1^ and 1550 cm^−1^ were accredited to the amide I (C=O stretched) stretching vibration and the amide II (C–N stretched and N–H bend) in proteins of the stratum corneum, respectively. In the stratum corneum, fluidization of the lipid bilayers and structural changes resulted in a shifting of CH_2_ stretching vibration to the elevated wave number. In the cases of F1, F5, and F9, the peaks of the epidermis at 3300 cm^−1^ shifted to higher wave numbers, as shown in Figure 6. In case of the F9 formulation, this shift of major peaks is significant and depicts that the formulated transdermal patches affect mainly the protein and lipids of the stratum corneum, which results in a greater amount of drug permeation followed by a greater amount of drug retention. The ATR-FTIR results of the dermis depicted that fluidization of the skin occurred at 3300–3380 cm^−1^ (representing OH–NH) and 2850–2930 cm^−1^ (representing asymmetrical CH_2_ skin lipids, keratin, and ceramide) when formulations F1, F5, and F9 were applied (Figure 6). Both the epidermis and dermis were affected by the patch formulations, ultimately resulting in the increased permeation and retention of the drug.

### 3.10. In Vivo Studies

Table 1 summarizes the parameters of optimized methotrexate-loaded patch formulation. The optimized patch formulation of methotrexate had a plasma level of 177.38 ± 4.7 ng/mL per drug plasma concentrations. The improved permeability of methotrexate in typical applications is useful for the determination of drug penetration to the systemic circulations with the scenario of methotrexate-loaded patch application. The clinical efficacy of methotrexate is somewhat determined by the systemic pathways of diseases, such as psoriasis; topically applied medication has been proven to be more beneficial. Low serum residual concentration was observed in case of methotrexate-loaded transdermal patches, and the accumulation of the drug in the major system minimizes the associated toxicities. The biological half-life (t_1/2_) calculated from the transdermal use of methotrexate increased in rabbits for 4–10 h^−1^ to 17.3 ± 2.21 h^−1^, showing statistically significant difference (*p* < 0.05). The methotrexate transdermal patches remain for a longer time period, producing sustained, controlled effect. Methotrexate has a substantially lower elimination constant (0.041 ± 0.0003 h^−1^) and an extensive mean residence time (MRT) (12.86 ± 0.59 h). These characteristics support extended activity of the drug in the patch formulations.

The improved bioavailability of medications is also depicted under the increased AUC values achieved with patches formulations. This may be associated to the hepatic first-pass effect being bypassed and the avoidance of stomach degradation. The drug retention data derived from the skin layers (epidermis and dermis) after in vivo studies showed statistically significant difference (*p* < 0.001) when the optimized formulation was compared with the control formulation (methotrexate solution), as shown in the Figure 7.

## 4. Conclusions

This study concluded that transdermal patches can be selected as an ideal choice for skin-delivery-related problems. In psoriasis, the barrier nature, i.e., stratum corneum, grows more rigid, and delivery of the medication becomes more challenging. Our study concluded by offering polymer potentials and resolving different shortcomings of topical drug delivery for the management of psoriasis. Anti-psoriatic medication was used for the evaluation of effectiveness, followed by better penetration and enhanced dermatokinetics.Topical administration therapy of formulated transdermal patches of methotrexate containing CMC-Na and HPMC polymers at different concentrations for the treatment of psoriasis were studied. All formulated transdermal patches (F1–F9) showed good physicochemical properties. In vitro release data concluded that polymer types, their concentrations, and their combination ratios significantly affected drug release from the delivery system, and enhanced drug release was exhibited by CMC-Na at higher concentrations. CMC-Na at a high concentration mainly affected proteins (ceramides and proteins) of the skin, resulting in higher penetration and retention of methotrexate in the skin. The formulated transdermal patches had the best properties used in the topical administration of methotrexate with enhanced penetration of skin and retention, providing a capable alternative for oral methotrexate tablets. The topical use of formulated transdermal patches demonstrated that reduced serum concentrations were achieved, which results in better patient compliance while reducing systemic toxicities.

## Figures and Tables

**Figure 1 polymers-14-01310-f001:**
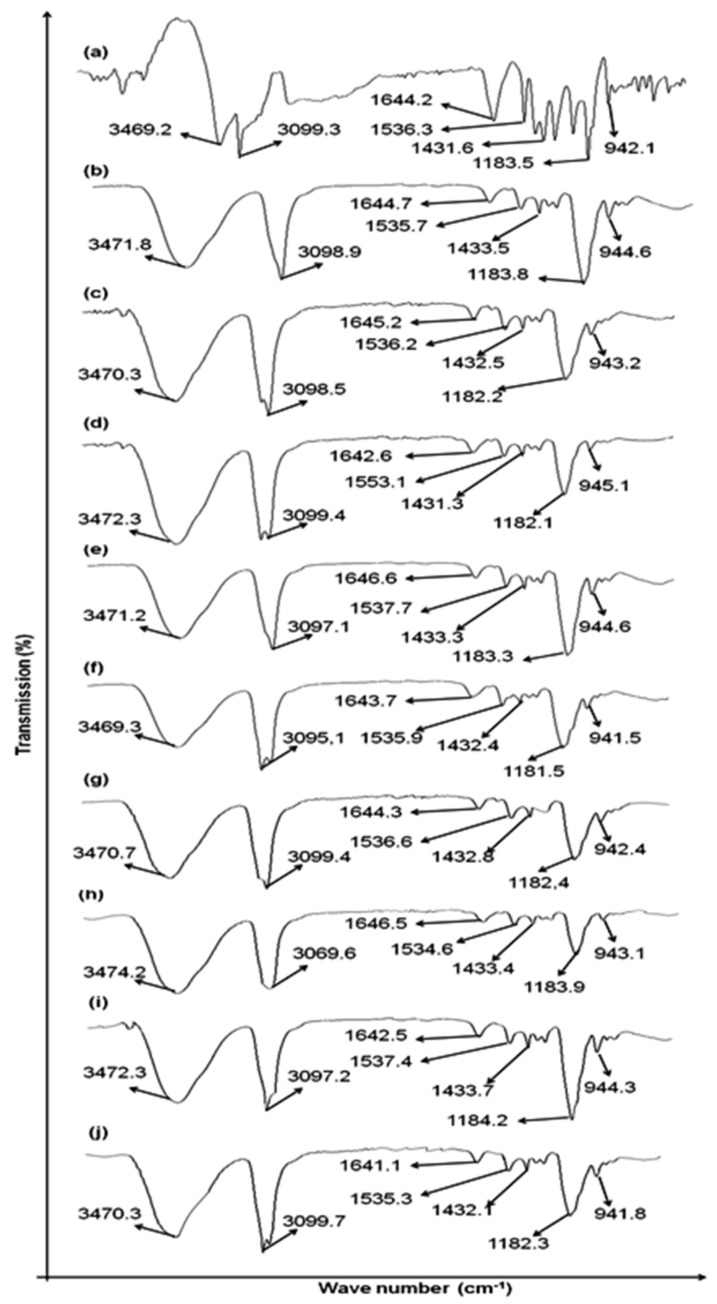
ATR-FTIR spectra of (**a**) pure drug methotrexate, (**b**) F1, (**c**) F2, (**d**) F3, (**e**) F4, (**f**) F5, (**g**) F6, (**h**) F7, (**i**) F8, and (**j**) F9.

**Figure 2 polymers-14-01310-f002:**
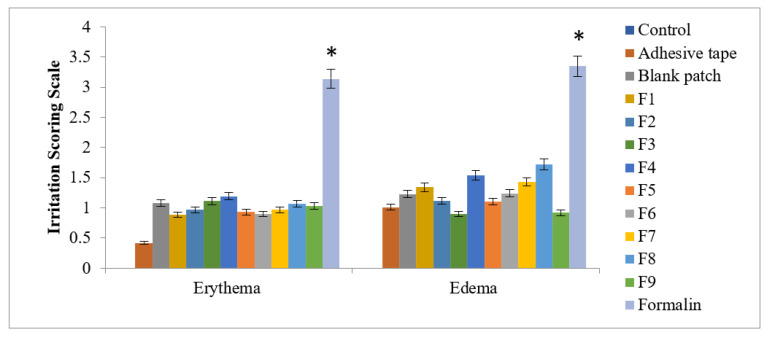
Skin irritation study: edema and erythema. Data are expressed as mean ± SD; *n* = 3. The results were significant compared to formalin (one-way ANOVA followed by post hoc Tukey test; *p* < 0.05).

**Figure 3 polymers-14-01310-f003:**
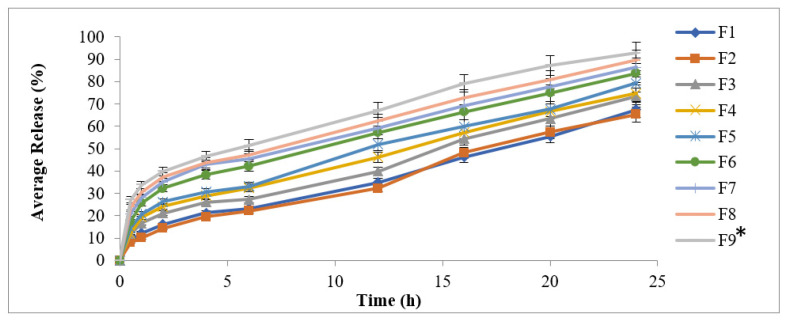
Release profiles of methotrexate from formulated patches (F1–F9). Data are expressed as mean ± SD; *n* = 3. One-way ANOVA followed by post hoc Tukey test (*p* < 0.05), F9 vs. F1.

**Figure 4 polymers-14-01310-f004:**
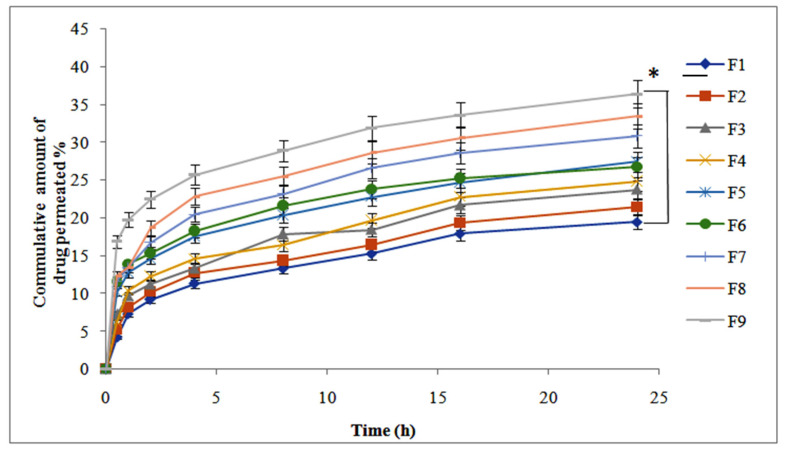
Permeation profiles of methotrexate from formulated patches (F1–F9). Data are expressed as mean ± SD; *n* = 3. One-way ANOVA followed by post hoc Tukey test (*p* < 0.05), F9 vs. F1.

**Figure 5 polymers-14-01310-f005:**
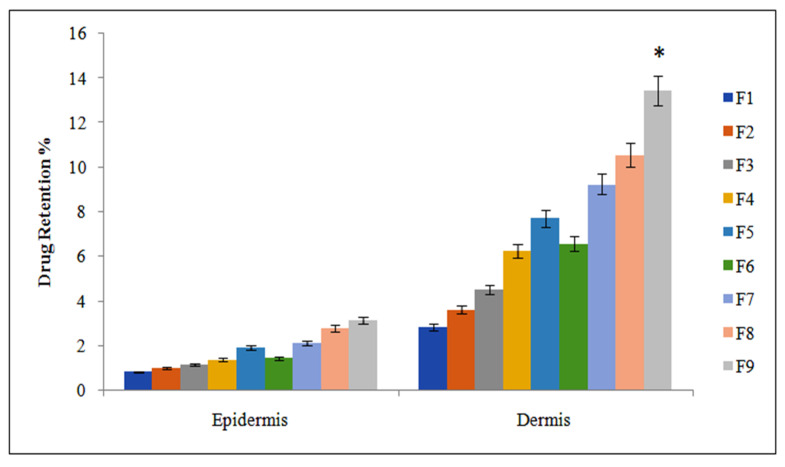
Skin drug retention analysis of methotrexate patches (F1–F9). Dataare expressed as mean ± SD; *n* = 3. One-way ANOVA followed by post hoc Tukey test (*p* < 0.05), F9 vs. F1.

**Figure 6 polymers-14-01310-f006:**
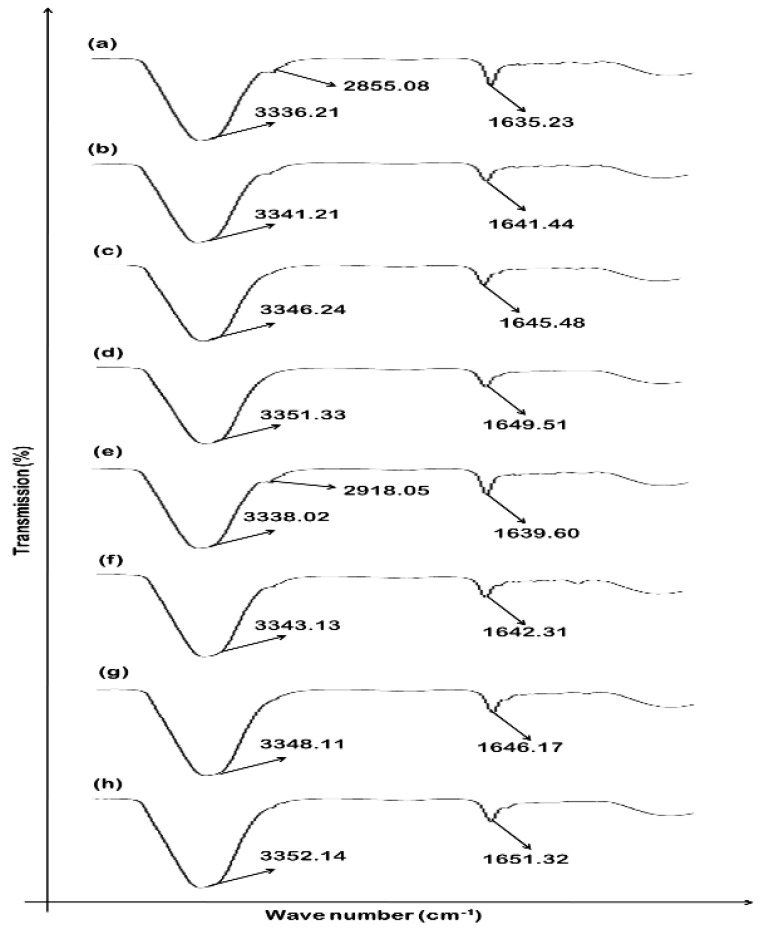
Representation of rabbit skin FTIR: (**a**) epidermis untreated, (**b**) epidermis treated with F1, (**c**) epidermis treated with F5, (**d**) epidermis treated with F9, (**e**) dermis untreated (**f**) dermis treated with F1, (**g**) dermis treated with F5, and (**h)** dermis treated with F9. Data are expressed as mean ± SD; *n* = 3.

**Figure 7 polymers-14-01310-f007:**
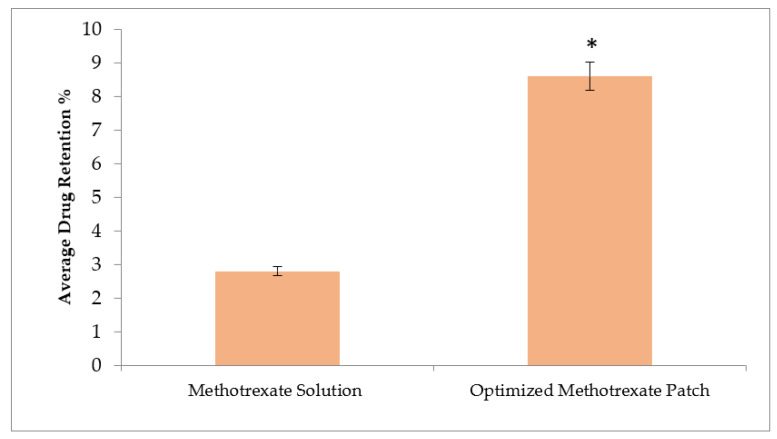
Methotrexate concentration in the skin from the methotrexate solution and methotrexate-optimized patch formulation (F9). Data are expressed as mean ± SD; *n* = 3. One-way ANOVA followed by post hoc Tukey Test (*p* < 0.05).

**Table 3 polymers-14-01310-t003:** Stability study of transdermal patches (F1–F9) at various temperatures and humidity levels.

F Code	Initial % Drug Content	25 ± 2C (60 ± 5%RH)			40 ± 2C (75 ± 5%RH)		
		15 Days	30 Days	60 Days	15 Days	30 Days	60 Days
F1	95.75 ± 4.22	95.54 ± 3.65	95.23 ± 3.56	94.87 ± 4.43	95.65 ± 4.55	94.32 ± 3.87	92.10 ± 4.47
F2	96.80 ± 3.32	96.45 ± 2.44	95.84 ± 2.43	95.34 ± 3.35	96.30 ± 3.43	95.67 ± 4.45	94.84 ± 2.65
F3	93.57 ± 5.34	93.26 ± 3.34	92.91 ± 4.76	99.42 ± 1.67	93.53 ± 3.26	93.05 ± 4.34	92.56 ± 3.58
F4	97.50 ± 3.21	96.76 ± 4.65	96.41 ± 3.46	96.11 ± 2.65	97.23 ± 2.43	96.75 ± 3.24	96.18 ± 4.62
F5	94.05 ± 4.23	93.69 ± 3.45	93.21 ± 4.81	92.79 ± 5.45	93.66 ± 4.54	93.34 ± 2.31	92.88 ± 2.16
F6	97.12 ± 2.43	96.77 ± 2.34	96.20 ± 2.13	95.14 ± 4.34	96.91 ± 3.12	95.64 ± 4.34	95.21 ± 1.63
F7	98.09 ± 1.56	97.67 ± 1.54	97.32 ± 1.26	96.77 ± 3.22	98.02 ± 1.21	97.87 ± 1.32	97.36 ± 3.73
F8	97.39 ± 2.35	97.08 ± 2.23	96.88 ± 3.25	96.65 ± 4.34	97.16 ± 2.24	93.82 ± 2.37	93.46 ± 1.48
F9	96.23 ± 3.32	95.74 ± 1.12	95.43 ± 1.22	94.52 ± 3.21	96.04 ± 1.43	95.64 ± 1.46	95.12 ± 1.52

Data are expressed as mean ± SD; *n* = 3; *p* > 0.05.

**Table 4 polymers-14-01310-t004:** Regression parameters of various formulations after fitting the drug release data to various release kinetic models.

F Code	Zero-Order		1st-Order		Higuchi		Hixon–Crowell		Korsmeyer–Peppas		
	K ± SD	R^2^	K ± SD	R^2^	K ± SD	R^2^	K ± SD	R^2^	K ± SD	R^2^	*n*
F1	6.083 ± 1.028	0.956	0.079 ± 0.121	0.932	5.661 ± 1.327	0.956	0.111 ± 0.095	0.942	2.578 ± 0.003	0.945	0.583
F2	5.931 ± 1.125	0.945	0.074 ± 0.124	0.928	5.523 ± 1.424	0.945	0.104 ± 0.100	0.935	3.389 ± 0.896	0.926	0.693
F3	7.094 ± 0.313	0.952	0.098 ± 0.107	0.932	6.392 ± 0.809	0.952	0.013 ± 0.077	0.942	1.865 ± 1.035	0.964	0.734
F4	7.518 ± 0.013	0.955	0.109 ± 0.099	0.945	6.639 ± 0.635	0.955	0.150 ± 0.068	0.954	3.648 ± 1.462	0.978	0.843
F5	7.991 ± 0.320	0.956	0.121 ± 0.091	0.942	7.043 ± 0.349	0.956	0.164 ± 0.057	0.953	1.564 ± 2.865	0.923	0.648
F6	8.952 ± 1.004	0.951	0.148 ± 0.072	0.950	7.676 ± 0.097	0.950	0.197 ± 0.034	0.960	2.685 ± 3.865	0.895	0.567
F7	9.413 ± 1.326	0.946	0.164 ± 0.060	0.949	7.995 ± 0.323	0.946	0.216 ± 0.021	0.961	3.234 ± 1.145	0.923	0.634
F8	9.867 ± 1.677	0.941	0.181 ± 0.048	0.942	8.337 ± 0.565	0.941	0.235 ± 0.008	0.958	2.647 ± 2.235	0.914	0.589
F9	10.495 ± 2.09	0.926	0.209 ± 0.028	0.938	8.783 ± 0.880	0.926	0.263 ± 0.012	0.953	1.564 ± 1.852	0.895	0.517

Data are expressed as mean ± SD; *n* = 3.

**Table 5 polymers-14-01310-t005:** Pharmacokinetic analysis of the optimized formulation (F9).

SNo	Parameters	Methotrexate-Optimized Formulation
1	Cmax (ng/mL)	177.38 ± 4.7
2	Tmax (h)	12
3	K (h^−1^)	0.041 ± 0.03
4	t1/2	17.3 ± 2.21
5	AUC0-24 (ng/mL.h)	2856.51 ± 123.2
6	MRT	12.56 ± 0.59

Cmax: Peak plasma drug concentration: Tmax—time at which Cmax was observed; K (h^−1^)—elimination rate constant; t_1/2_—elimination half-life; AUC_0–t_—area under the plasma concentration–time plot from 0 h to24 h. MRT: mean residence time, (*p* < 0.05 for t = 1/2). Data are expressed as mean ± SD; *n* = 3.

## Data Availability

Not applicable.

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
