# Peer review of "Formulation and Evaluation of Hydrophilic Polymer Based Methotrexate Patches: In Vitro and In Vivo Characterization"

_polymers, 2022, doi:10.3390/polym14071310_

Round 1
Reviewer 1 Report
The manuscript entitled “Formulation and Evaluation of Hydrophilic Polymers Based Methotrexate Patches: In Vitro and In Vivo Characterization” contains interesting findings and is suitable for publication. However, the following issues should be addressed, which are shown below:
- Abstract: The abstract should be improved by including the results obtained from the study.
- There are several grammatical errors, and punctuation issues to be corrected in the entire manuscript. I have indicated just a few examples shown below in the introduction section.
- Introduction:
- Correct as shown by including a comma and the word, “the”: In human body, the skin…
- Correct as shown: The rigid layer of skin is stratum corneum and it is tougher more than the horn of an animal.
- Psoriasis is a chronic autoimmune skin and joint…
- Methotrexate causes inhibition of dihydrofolate reductase enzyme and results in the inhibition of cell proliferation.
- The formulation of transdermal patches of methotrexate should be revised in terms of the solvents that were used to dissolve CMC ̶Na & HPMC.
MTX is not soluble in water. The authors indicated that MTX was dissolved in water. The statement should be corrected because it indicates that MTX was dispersed in water.
- The procedure for the evaluation of the Swelling Index should be revised. It is not clear.
- The unit should be stated for Water Transmission Rate in Table 2.
- The WVTR values of the patches were low and should be discussed.
- The significance of the results should be discussed. Presenting the results without discussing their significance is not good enough.
- Na2HPO4 should be well written.
Author Response
Step wise Reply to the comments of Reviewer#1
Greetings of the day Dear Reviewer!
First of all I am really indebted to you for your kind consideration of our research article. The valuable suggestions from your kind self regarding our research article reflected your thorough grounding and expertise in the specialized filed for which I am much grateful to you.
Please find below stepwise reply for your suggested comments.
Abstract: The abstract should be improved by including the results obtained from the study.
Reply: Thanks for suggestion. The abstract is improved after incorporating the results of important physicochemical tests to make it easier for the researchers to understand.
There are several grammatical errors, and punctuation issues to be corrected in the entire manuscript. I have indicated just a few examples shown below in the introduction section.
Introduction:
Correct as shown by including a comma and the word, “the”: In human body, the skin…
Correct as shown: The rigid layer of skin is stratum corneum and it is tougher more than the horn of an animal.
Psoriasis is a chronic autoimmune skin and joint…
Methotrexate causes inhibition of dihydrofolate reductase enzyme and results in the inhibition of cell proliferation.
Reply: Corrections have been made properly regarding grammatical mistakes throughout the manuscript.
The formulation of transdermal patches of methotrexate should be revised in terms of the solvents that were used to dissolve CMC ̶Na & HPMC.
Reply: Revised as suggested by specifying appropriate solvent system as shown in the Table-1 of manuscript properly highlighted by track change function.
MTX is not soluble in water. The authors indicated that MTX was dissolved in water. The statement should be corrected because it indicates that MTX was dispersed in water.
Reply: Correction made with proper citation in the manuscript. The literature depicted a peculiar behavior of methotrexate by stating that it is hydro-soluble at physiological pH because at pH 7.4 it exists in ionized form. So, on this analogy phosphate buffer solution pH 7.4 was used to dissolve the methotrexate.
The procedure for the evaluation of the Swelling Index should be revised. It is not clear.
Reply: Thanks for suggestions. In the manuscript the complete procedure for swelling index determination is incorporated that can be shown highlighted due to track function change.
The unit should be stated for Water Transmission Rate in Table 2.
Reply: Unit for water transmission rate is incorporated in the respective Table 2.
The WVTR values of the patches were low and should be discussed.
Reply: Thanks for suggestion. The WVTR values was discussed properly in the manuscript.
The significance of the results should be discussed. Presenting the results without discussing their significance is not good enough.
Reply: In the result and discussion section the significance of each result parameter is specified to highlight its importance.
Na2HPO4 should be well written.
Reply: Thanks for suggestion. Correction made accordingly.

Reviewer 2 Report
Paper titled (Formulation and Evaluation of Hydrophilic Polymers Based 2
Methotrexate Patches: In Vitro and In Vivo Characterization ) by Latif et al. is an interesting article testing theuse of patch formulas of methotrexate in comparison to solution formula.
1- Please highlight the rational of selection of the best formula in a separate paragraph. Which one was tested in vivo and why? what were the characters led to selection?
2- highlight this also in abstract and discussion.
3- In all illustration, mention in figure legends or table footnotes, what is the type of the presented data (mean, median, mode, ...etc).
4- Write a section on statistical analysis in methods
5- Mention stat test used in any figure
6- Introduction is too long and need to be shortened to be more concrete
Author Response
Dear Reviewer!
I acknowledge your efforts regarding sparing your precious time to evaluate our research article and gave valuable assessment to make it publishable. Yours comments are commendable and helped us a lot to refine this manuscript as well as will help us in future to keep in mind all these parameters to make the research articles acceptable in each and every aspect. After proper thanking you please find below the step wise answers of your suggested comments.
Paper titled (Formulation and Evaluation of Hydrophilic Polymers Based 2
Methotrexate Patches: In Vitro and In Vivo Characterization ) by Latif et al. is an interesting article testing theuse of patch formulas of methotrexate in comparison to solution formula.
1- Please highlight the rational of selection of the best formula in a separate paragraph. Which one was tested in vivo and why? what were the characters led to selection?
Reply: Formulation optimization is crucial parameter to achieve best possible formulation that could be used for further studies. Our study comprised of 09 formulations, among which formulation F9 was labeled as our optimized formulation owing to best physicochemical properties, in-vitro drug release, ex-vivo drug permeation and skin retention results. Thus F9 formulation was further selected for in-vivo studies.
2- Highlight this also in abstract and discussion.
Reply: Thanks for suggestions. Highlighted in abstract and discussion section.
3- In all illustration, mention in figure legends or table footnotes, what is the type of the presented data (mean, median, mode etc)
Reply: Thanks for suggestion. Added data type in the figure legends and table footnotes.
4- Write a section on statistical analysis in methods.
Reply: Various statistical tests were applied in our study depending upon nature of the data as and when necessary including student t-test, One and Two way ANNOVA. These tests are properly placed in the mythology section also.
5- Mention stat test used in any figure.
Reply: Thanks for suggestion. Specific tests mentioned in the figures.
6- Introduction is too long and need to be shortened to be more concrete.
Reply: Thanks for suggestion. Necessary corrections made to refine the introduction part.

Round 2
Reviewer 1 Report
The authors have addressed the issues raised. However, there are several spelling errors in the entire manuscript to be carefully corrected by the authors. I have listed some of such errors:
Figure 7: ...formulation (F9) respectively calculated from form in-vivo studies...
Conclusion: ...delivery of medication becomes more challenging changeling...
...at different concentrations for the treatment of psoriasis...
...significantly affect drug release from the delivery system...
...making it capable alterations alternative to the oral...
Author Response
Dear Reviewer!
I acknowledge your efforts regarding sparing your precious time to evaluate our research article and gave valuable assessment to make it publishable. Yours comments are commendable and helped us a lot to refine this manuscript as well as will help us in future to keep in mind all these parameters to make the research articles acceptable in each and every aspect. After proper thanking you it’s to bring into your kind attention that the entire manuscript has been thoroughly checked and all the spelling errors have been corrected carefully.
The authors have addressed the issues raised. However, there are several spelling errors in the entire manuscript to be carefully corrected by the authors. I have listed some of such errors:
Figure 7: ...formulation (F9) respectively calculated from form in-vivo studies...
Conclusion: ...delivery of medication becomes more challenging changeling...
...at different concentrations for the treatment of psoriasis...
...significantly affect drug release from the delivery system...
...making it capable alterations alternative to the oral...
Response: All the corrections were made accordingly throughout the manuscript.

Reviewer 2 Report
Although the manuscript was partly improved I found the following is mandatory to be clear before any further processing:
1- Authors wrote (2.13. Statistical Analysis 399 The statistical tool used in this study was student t-test, one-way ANNOVA and two 400 way ANNOVA using SPSS software version 16. ), this is completely not clear and does not make sense
2- Authors have to check the normality of distribution of the results by a suitable post hoc test (such as Shapiro-Wilk test or K-S test) before deciding to choose certain ANOVA. If the normality test indicated normal dist of the data, so use one-way ANOVA, if not, use non parametric ANOVA. In all cases choose a suitable post-hoc test
Modify methods & results in this light.
3- Authors should give the source of chemicals, kits and antibodies completely and consistently (code, company, town, state and country) & version for software.
4-Authors should mention the Table footnotes (Not title) the type of data
5- Authors did not mention with each illustartion what was the type of analysis applied to these data.
6- Data in Figure 2,4,5,7 lacks statistical analysis
Author Response
Greetings of the day Dear Reviewer!
First of all I am really indebted to you for your kind consideration of our research article. The valuable suggestions from your kind self regarding our research article reflected your thorough grounding and expertise in the specialized filed for which I am much grateful to you.
Please find below stepwise reply for your suggested comments.:
- Authors wrote (2.13. Statistical Analysis 399 The statistical tool used in this study was student t-test, one-way ANNOVA and two 400 way ANNOVA using SPSS software version 16. ), this is completely not clear and does not make sense
Reply: Since, Data of this study are having different variables, so depending upon suitability and feasibility of statistical test, we have applied student-t test and One Way ANNOVA along with mean ± SD (n=3).
- Authors have to check the normality of distribution of the results by a suitable post hoc test (such as Shapiro-Wilk test or K-S test) before deciding to choose certain ANOVA. If the normality test indicated normal dist of the data, so use one-way ANOVA, if not, use non parametric ANOVA. In all cases choose a suitable post-hoc test.
Reply: We have checked data normality by graph pad prism software. This software suggested One Way ANNOVA (Dunnet Test). That’s why we use One Way ANNOVA.
Modify methods & results in this light.
- Authors should give the source of chemicals, kits and antibodies completely and consistently (code, company, town, state and country) & version for software.
Reply: The code, company, town, state and country of chemicals have already been specified in the methodology portion of manuscript.
4-Authors should mention the Table footnotes (Not title) the type of data.
Reply: The footnotes of all the tables have been incorporated as specified by your kind self.
- Authors did not mention with each illustration what was the type of analysis applied to these data.
Reply: The type of analysis has been specified to each illustration in the manuscript as per your guidance.
- Data in Figure 2,4,5,7 lacks statistical analysis.
Reply: Statistical analysis have been incorporated in the above mentioned figures (2, 4, 5 & 7).

Round 3
Reviewer 2 Report
Authors improved the manuscript partly: I have 2 essential criteria without them, this paper cannot be accepted for publication.
Authors should confirm they consulted a statistician who revised their work and approved the data presentation and stat analysis test:
1- For the 2 groups comparison such as figure 7: no significance sympols are there despite that in the text, authors wrote in the text that there is a significant difference between the groups.'
2- ANOVA should be followed by a post hoc test to compare between the individual groups.
Author Response
Greetings of the day Dear Reviewer!
Hope you will be doing well.
Thanks for reviewing the manuscript and suggested valuable corrections. Statistical analysis portion was rechecked with the help of experts from the Department of Statistics, Gomal University and add relevant tests in the methodology section as well as in the figures. For comparison between groups ANOVA followed by Post hoc Tukey test was used.
Please find below stepwise reply for your suggested comments.:
- For the 2 groups comparison such as figure 7: no significance sympols are there despite that in the text, authors wrote in the text that there is a significant difference between the groups.'
Response: Thanks for comments. The significant symbols incorporated in all the figures.
- ANOVA should be followed by a post hoc test to compare between the individual groups.
Response: Thanks for guidance. ANOVA test was followed by post hoc Tukey honest significance test in all comparison groups.
